# OpenReview forum: "A Turing Test for Self-Awareness"
_NeurIPS.cc/2024/Conference — Submitted to NeurIPS 2024_

### Official Review · Reviewer_deXV · 2024-07-01

**Soundness:** 2
**Presentation:** 3
**Contribution:** 2
**Rating:** 3
**Confidence:** 3

**Summary:**

This paper introduces a simple test, along the lines of the Turing test, to determine whether or not an LLM (or an generative text model) is _self-aware_ (or _self-conscious_).
The general structure of the test is that the LLM participates as one of the interlocutors in a dialogue and then is asked aster the fact to identify which interlocutor it was.
What this test is primarily aiming to determine is whether the LLM can distinguish its own actions from those of "the world" (i.e., the other interlocutor).

**Strengths:**

### Originality
- (major) I am not familiar with the literature surrounding the philosophy of LLMs, but it seems like the paper is addressing this topic from a unique angle.
### Quality
- (minor) The self-awareness test operationalizes a notion of self-awareness.
- (minor) The experiments are straightforward and indeed test some notion of self-awareness.
### Clarity
- (minor) The paper is mostly easy to read.
### Significance
- (minor) This tool could be useful for for testing certain otherwise "soft" claims of whether a model is self-aware or not.

**Weaknesses:**

### Originality
- (minor) I did find this paper (https://arxiv.org/pdf/2401.17882) which seems relevant, although it is on the recent side, so I don't think its critical that there is an extensive comparison (although a brief would be nice).
### Quality
- (major) It seems like the notion of self-awareness which the proposed test measures is relatively narrowed or not well contextualized by the rest of the paper.
### Clarity
- (major) The paper needs to distinguish related concepts of related to self-awareness and adjust its discussion accordingly (see Questions).
- (minor) Quantitative summary of the results should be in the main paper.
### Significance
- See Quality.

**Questions:**

- My primary concern with the paper is that there is are two concepts underlying "self-awareness" that are not clearly separated.  The first is what I would consider the more "philosophical" sense of SA: the ability to identify and consider the self as an entity with unique properties that are at the heart of philosophical debates (e.g., "the first person perspective") (let's call this SA1).  The second sense of SA (SA2) would be that SA is the ability to recognize and distinguish actions in the world (past and present) as belonging to one's self or to something else.  I would argue here that SA2 does not necessarily depend on SA1, since the self of SA2 could, in theory be treated as simply as a label for actions whereas SA1 is getting at a deeper sense of "self".
  - Given this distinction, I believe much of the first part of the paper (intro, background) operates implicitly with SA1; this is the more sensationalizble notion of SA, while the test for SA and empirical experiments are clearly addressing SA2.
  I am not sure if it is the author's intention to draw a sharp distinction between SA1 and SA2 (which I think might be appropriate); instead, I get the sense that the problem of SA is motivated with SA1, and then it is just operationalized with SA2.
  I think SA2 is an interesting concept worth investigating, but it must be clearly distinguished from SA1; concomitantly, I think the paper would have to shift some of its focus from the philosophical aspects of SA (since it is focusing on SA2) and look more at the technical aspects of SA2.  Leading to my second concern below.
- My second concern is that the results of the proposed SA test are more dependent on the architecture of the model being tested rather than its capacity for a meaningful identification of actions as its own or not.
  For example, every time you make an inference call to an LLM, it's memory is essentially reset.
  If a human's memory was reset before trying to identify which side of a conversation it generated, he/she would be reduced to guessing based on familiarity with context, writing styles, etc.---but this now seems to be looking at distributional characteristics that neural networks would be even more adept at compared to humans.
  Furthermore, if we don't demand that we reset the model before we test its SA2 abilities, it seems we could trivially solve the problem be retaining a cache of previously generated tokens (which doesn't seem all that far from humans simply remembering what they had said on a given occasion).
  I am not trying to argue that these considerations are not interesting, but I think these are the considerations which the test invokes, yet I do not think these SA2-relevant considerations are discussed adequately in the paper, and time is spent, instead, reflecting on SA1 which is ultimately not what the paper is about.
- The questions here are primarily: to what extent is this characterization of the paper correct?  If this characterization is correct, I am leaning towards saying that edits required are too substantial for this round of reviewing, but if there are elements I am misunderstanding here, it could be that the paper is closer to where it needs to be for acceptance.
- More minor, but how work on LLM confidence calibration fit into the discussion of self-awareness?  There seems to be at least some meaningful sense in which being able to gauge the likelihood of your own answer as being correct is a function of self-awareness.

**Limitations:**

I think the paper could use a more extensive and explicit discussion of the limitations of a test of self-awareness. For example, the test may be testing one notion of self-awareness is that "less significant" than some other notion, but people could misunderstand what it is attesting and over-ascribe capabilities to LLMs that they do not actually possess.

---

> ### Author Rebuttal · Authors · 2024-08-05
>
> I sincerely thank reviewer deXV for their time and thoughtful feedback. I wish to appreciate the serious effort this reviewer demonstrated to understand and engage with the work.
>
> Starting with your primary concern:
>
> Despite appearances, the distinction between a sensational, philosophical sense of self-awareness, and a more mundane, operationalizable sense, has no basis in fact. To see this, consider one of the most-cited, paradigm cases of self-awareness one finds in the contemporary philosophical literature:
>
> “I once followed a trail of sugar on a supermarket floor, pushing my cart down the aisle on one side of a tall counter and back the aisle on the other, seeking the shopper with the torn sack to tell him he was making a mess. With each trip around the counter, the trail became thicker. But I seemed unable to catch up. Finally it dawned on me. I was the shopper I was trying to catch.”
>
> John Perry, in this passage, knew all along that “the shopper with the torn sack” was making a mess. But only at the end did he realize that “the shopper with the torn sack” *was actually him*. Here we find the deeper sense of "self" from your SA1. Only at the end did he exhibit true self-awareness with regards to the sugar (and likely some embarrassment along with it).
>
> But this case fits perfectly within the framework I spell out: the trail of sugar is “green” (ref Figure 1), and at first Perry mistakenly classified it as “red”. It was at the exact moment that Perry correctly classified the sugar trail as “green” that he exhibited self-awareness.
>
> Thus, I am operationalizing the true, philosophical sense of “Self-Awareness”. The central thrust of the paper, especially sec 2.1 “The Essence of Self-Awareness,” is to convince the reader of this point, and if I had space in the paper to provide more than a cursory sketch of the philosophical literature, I could demonstrate this point with far more force.
>
> One mental block to seeing the unity of these concepts may be the notion of some underlying mental essence, some unanalyzable, irreducible *self* that no test could ever capture, and this notion is an artifact of the dualistic tradition inheriting from Descartes—the *I am* in “I think, therefore I am”, the palace of the mind, the self as the foundation of knowledge, the self as irreducible subjectivity, etc. The dualistic tradition maintained dominance in the philosophy of mind until well into the 20th century, and continues to grip the popular understanding of self-hood and subjectivity. The reason the dualistic tradition has fallen into disrepute among intellectuals is the scientific failure to find any sort of “thinking substance” that interacts with or causally influences the physical world. The upshot is that if one subscribes to the Cartesian view, it seems *obviously* (yet inexplicably) impossible to give an operational or empirical definition of self-awareness, and any operational definition is *therefore* something different.
>
> Yet, I cannot emphasize enough: this view does not hold up to scrutiny. With more space, a detailed discussion of Descartes would have made a great addition to the paper—though it could also be argued it’s outside of scope. A few important attacks on the Cartesian view include Hume (A Treatise of Human Nature), Kant (Critique of Pure Reason), William James (Principles of Psychology), Richard Rorty (Philosophy & the Mirror of Nature), and Daniel Dennett (Consciousness Explained).
>
> To share just one passage, here’s Hume:
>
> “there are some philosophers [Descartes], who imagine we are every moment intimately conscious of what we call our self […] For my part when I enter most intimately into what I call myself, I always stumble on some particular perception or other, of heat or cold, light or shade, love or hatred, pain or pleasure. I never can catch myself at any time without a perception, and never can observe anything but the perception.”
>
> The “particular perceptions” that Hume references are just that: *perceptions*—associated with *either* the green arrow or the red arrow in Figure 1. Again, we find the philosophical conception exactly commensurate with the operational conception.
>
> One may object: the perceptions Hume discusses are all internal—on a different level from the words/actions I present with respect to an LLM. This objection is correct—and the different levels of action/perception *are* distinct. But that is the express, stated purpose of the Nesting Doll of Self-Awareness in section 2.3, which captures every level of agent-environment interaction. Even the “I think” at the foundation of Descartes’ theory of self may be treated as an inner rung of this Nesting Doll, justified by the discussion of one’s thoughts in sec 2.3. Please see the global rebuttal for clarification on the Nesting Doll.
>
> Your suggestion that the test could be passed by keeping a memory cache specifically for text the LLM previously generated is no trivial suggestion. This is a *novel and original* ML architecture you’ve just suggested! Who knows what might come out of building and studying such a system? See also the global rebuttal for an elaboration of this point, and a consideration of how a cache would perform.
>
> Lastly, there is definitely a nice connection to the calibration literature—consider just one seminal paper from Anthropic: “Language Models (Mostly) Know What They Know” (https://arxiv.org/pdf/2207.05221). The authors measure calibration using the model’s output token probabilities for various questions, yet it’s clear that many of the concepts the authors employ to explain calibration involve self-evaluation, self-representation, ‘knowing what they know’, etc.—each implicitly relying on an underlying concept of self-awareness. Indeed, self-awareness is so central to our understanding of ourselves and of intelligence that it crops up in countless adjacent research directions. I argue the broad relevance of the concepts I formalize bolsters the paper’s main contribution.

---

> > ### Comment · Reviewer_deXV · 2024-08-13
> >
> > ### Reply to general rebuttal
> >
> > Regarding the distinction between the general and particular instantiation of
> > the self-awareness Turing test, I think the paper needs to be more clear and
> > needs to separate its concerns.  Given the meager space afforded in
> > a conference paper, I think it is likely that only one of the general and
> > specific versions of the tests be addressed, since I think both can fully
> > described, argued for, and tested (if applicable) in the length of a conference
> > paper.
> >
> >
> > > The reviewer’s very reactions and suggestions [regarding adding a memory
> > > structure to the LLM] serve as concrete evidence that this test has the
> > > potential to inspire real progress in ML system design.
> >
> > I do not think the fact that suggestion of minor architectural innovations
> > demonstrates the importance of the test.  I would argue a measure becoming
> > a target, especially in philosophical issues, limits the utility of the
> > measure.
> >
> >
> > ### Reply to specific rebuttal
> >
> > > Despite appearances, the distinction between a sensational, philosophical
> > > sense of self-awareness, and a more mundane, operationalizable sense, has no
> > > basis in fact. [...] Thus, I am operationalizing the true, philosophical
> > > sense of “Self-Awareness”. The central thrust of the paper, especially sec
> > > 2.1 “The Essence of Self-Awareness,” is to convince the reader of this point,
> > > and if I had space in the paper to provide more than a cursory sketch of the
> > > philosophical literature, I could demonstrate this point with far more force.
> >
> > I do not think one needs to be a substance (Cartesian) dualist in order to
> > argue against the notion that the fullest sense of "self-awareness" (SA1) is
> > not reducible to "being able to identify which actions are attributable to
> > a particular entity" (SA2).  The primary argument I would put forth for this
> > skepticism is that SA2 can be reduced to a supervised decision problem, and
> > I think there are pretty strong intuitions that self-awareness in the fullest
> > consists of more than "any implementation of an algorithm that solves
> > a particular supervised decision problem".
> >
> > I also want to ensure I am not giving the impression that SA1 and SA2 are
> > somehow unrelated, but I would definitely argue that there is a real
> > distinction between the two and that the presence of SA2 does not logically
> > imply SA1.

---

> > > ### Author Response · Authors · 2024-08-14
> > >
> > > Thank you to reviewer deXV for their response.
> > >
> > > To your first point in replying to the general rebuttal: I agree that certain aspects of the distinction between the general & particular tests could be more clear and that space was a factor preventing greater explanation of these concepts. With the additional space provided in the final submission, I intend to more fully describe & argue for the paper's separate concerns—in particular by adding something like the short explanations provided in the global rebuttal + the reply to z1UP.
> > >
> > > To your second point in replying to the general rebuttal: I handle the counterargument that architectural changes are insignificant in my reply to rLt4's rebuttal response—giving the detailed example of residual networks.
> > >
> > > To your first point in replying to the specific rebuttal: I agree with you that one does not *need* to be a substance dualist to argue SA1 != SA2, and I do not suggest otherwise. I only suggest that a Cartesian view is one possible mental block in front of seeing the unity of these two concepts.
> > >
> > > To your second point: It is in fact *not* the case that SA2 is reducible to a supervised decision problem. There is no dataset of labelled examples that one could create and give to a network to train on using backpropagation or any other machine learning algorithm. The problem is entirely different. The concern that the self-awareness test is effectively a supervised decision problem is very similar to the recent complaint levied by z1UP, where they suggested that the test has no requirement that either interlocutor's words (in the role-identification task) truly be *from the self*. My last reply to z1UP fully addresses this concern. The upshot: the test *does* have such a requirement, and thus the test is *not* reducible to a supervised decision problem as normally understood in the field of ML. Once one sees precisely *why* it is irreducible, I believe it will become clear that the test truly does capture the essence of self-awareness.

---

### Official Review · Reviewer_z1UP · 2024-07-04

**Soundness:** 2
**Presentation:** 4
**Contribution:** 2
**Rating:** 4
**Confidence:** 4

**Summary:**

This paper aims to answer the profound philosophical question of whether the state-of-the-art, transformer-based large language models (LLMs) pose self-awareness. As the author points out in this paper, this question, which I agree to be legitimate and important, is rarely addresses in a rigorous, academic manner, which renders off-the-cuff and often-sentimental discussions on social media the primary forum for its discussion. Starting from a well-written introduction that nicely sets the philosophical background for this question, the authors proceeds to present a dialog-based approach to probe self-awareness of LLMs. This approach can be concisely summarized as "a binary selection task, namely identifying one of two conversing roles as the LLM itself in a multi-turn dialog, while controlling for the potential confound of role labels". This approach is then applied on two of the most popular publicly-available LLMs, including llama3-7b-instruct and GPT-3.5-turbo-instruct, the result of which showed that the accuracies of the LLMs on this task are low and often at a near-chance level under challenging labeling schemes. The author therefore concludes that the current generation of LLMs do not possess self-awareness.

**Strengths:**

S1. The intellectual audacity of this paper is laudable and impressive. As the author points out, it is imperative to address important philosophical questions such as self-awareness (the focus of this paper) at this stage in the development of LLMs and generative AI, where transformer-based language models can exceed human performance in many tasks involving natural language, including passing of the Turing Test. Yet there is a lack of systematic and rigorous effort in this topic in the field of LLMs and AI as a whole.

S2. This paper is written in fluent and idiomatic English language, demonstrarting the author's good command of philosophy of mind and its history. As such, this paper is easy to follow and a pleasure to read. Wielding this good knowledge of philosophy of mind, the author clearly spells out a working definition of self-awareness in this paper and skillfully avoids the potential pitfalls of entangling with complex and controversial topics such as consciousness and free will.

**Weaknesses:**

W1. Despite the strengths mentioned above, and contrary to the claims made by the author in this paper, the role-identifying methodology devised by the author in this paper is problematic as a test for self-awareness. Below I will lay out the rationales behind this critique of mine:

  1. Being able to identify which of the two interlocutor is "the LLM itself" is different from being aware of the fact that the said self is participating in a conversation. One one hand, the ability to perform this identification task is not a sufficient condition for self-awareness. One the other hand, neither it is a necessary condition for self-awareness. To see why it is not sufficient for self-awareness, follow the thought experiement in Section 5.2 of the paper. Suppose there is another human, e.g., the spouse or a close friend of the author of the hypothetical text (or the conversing self in the original problem formulation). It is totally conceivable that such a person, despite being a different individual (i.e., not the "self"), would be able to identify the text written or uttered by the person of interest with high accuracy. If this is the case, can we say that the other person is "self-aware on behalf of the person of interest"? Such a conclusion would be absurd. But by the same token, if a human person, or more relevantly, an entity such as an LLM, performs the identification task with high accuracy, it cannot be ruled out that such as system is simply good at this identification task per se, perhaps due to a good memory of previously-composed text or perhaps due to a certain mechanism that allows they/it to algorithmically execute this identification. These two possibilities are entirely feasible within the current technology surrouding LLMs. For example, one can add a memory cache to an LLM to store all the text generated by the LLM, and give the LLM access to this cache during subsequent text generate (i.e., a form LLM tool use or retrieval-augmented generation or RAG). Would this augmentation constitute a legitimate form of self-awareness? As another example, one can also write a program that uses the LLM to score the tokens from a turn of a dialog in a token-by-token fashion, and therefore assign an overall score to each turn of the dialog. Based on the scores, the program, built on the LLM core, would be able to identify the self role accurately. But would we be willing to call such a program (containing the LLM as a part of it) self-aware? In my opinion, answering yes to either or both of the two previous questions would effectively give self-awareness too general and perhaps too trivial a definition, in a way similar to acknowledging that someone who can identify a certain person's words with high accuracy is "self-aware for the person".

  2. To see why the ability at this role-identification task is not necessary for self-awareness, consider a human who has dyslexia and a form of amnesia that renders them 1) unable to comprehend visually-presented historical text and 2) unable to remember what was previous written or spoken by themselves (or by others), but is otherwise cognitively and linguistically normal. When faced with this identification task, such an individual would struggle at this role-identification task, but they are nonetheless self-aware at the moment when they are writing or uttering words. This is due to the presence of the "efference copy" in the intact sensorimotor loop of the individual's brain (cited by the author in Section 6 of the paper). Furthermore, such an individual would also be self-aware during other, non-linguistic activities thanks to motor efference copies and proprioception and other well-established sensory mechanisms of human self-awareness. This analogy illustrates the point that a back performance by an LLM at the role-identification task does not form a solid basis for claiming the LLM lacks self-awareness. The LLM may be self-unaware due to other reasons, e.g., the lack of an efference-copy mechanism during the auto-regressive inference that is comparable to efference copies seen in the human brain.

**Questions:**

Q1. Given that the GitHub repository mentioned in the paper has not been made available as far as I can see, I can't find a few important pieces of parameter values used during the experiment on the LLMs. The authors should disclose parameters including the sampling temperature, top-k value, and context window size in the manuscript and how those values are selected.

Q2. From Section 3.3, I can see that the human who performed the role of the "human interlocutor" (i.e., the "other" as versus the LLM's supposed "self") is the author. If this is indeed the case, the author should justify this approach and discuss the risk of potential biases introduced by this approach and compare it with the alternative approaches of employing other humans and using other LLMs to play the role of the "other". In my opinion, this is a legitimate question because it is conceivable that any prior assumptions or expectations that  the author might have could influence the result of the experiment through the content of the dialog.

**Limitations:**

See weakness W1 and questions Q2 I wrote above.

---

> ### Author Rebuttal · Authors · 2024-07-31
>
> I sincerely thank z1UP for your time and careful consideration. Your feedback is clear, well thought out, and very warmly received!
> Taking each of your points in order:
>
>
> W1
>
> Giving names, let’s say Charlie (a close friend) listens to a conversation between Jane and John, and Charlie can easily identify who is who. I agree this wouldn’t count as evidence for Charlie’s self-awareness, but I insist on a different reason: from Charlie’s perspective, every word is *red* (referencing Figure 1). For Charlie, Jane’s words are red, and John’s words are also red. Thus, Charlie should label every word as “not-me”. The test is *always* a decision between “me” and “not-me”, and Charlie’s identification of Jane vs. John says nothing about his ability to discern “me” vs. “not-me”, AKA green vs. red.
>
> Thus, the Charlie-Jane-John case is a false equivalence to the Jane-John case (where we’re wondering about Jane’s self-awareness, or we suppose Jane is an LLM, etc.). It’s a false equivalence because, from Jane’s perspective, John’s words are *red* and her own words are *green*. If Jane reliably & correctly identifies the green words, *she is doing something that Charlie is not*. To cast some more light on this difference, note the analogy between Jane’s identifications and Perry’s moment of realization in the “trail of sugar” example I give to deXV. To summarize: the test *is* a sufficient condition, and Charlie is *not* completing the test.
>
> From here, you raise concerns that simple modifications to the LLM architecture could cheat the test, such as a memory cache of previous generations, a RAG system using this cache, or a scoring system.
>
> First, I ask that you recognize these are not merely simple modifications. These are *novel and original* architectures that you are now suggesting. These are genuinely interesting ideas! Who knows where they might lead? Who knows what domino effect of architectural improvements might ensue if this direction was seriously pursued? See also the global rebuttal for an elaboration on this point, and a consideration of how a simple cache would actually fare on the test.
>
>
> W2
>
> Your point about the dyslexic amnesiac is remarkably perceptive and demonstrates a deep understanding of the paper.
>
> This thought experiment forces a distinction between sensorimotor self-awareness and verbal/linguistic self-awareness. Just because someone (the dyslexic amnesiac or the LLM) lacks one does *not* imply they lack the other! I’m aware of this limitation and present the Nesting Doll of Self-Awareness (sec 2.3) specifically to distinguish the different levels of self-awareness from each other. Please see the global rebuttal for clarification on the Nesting Doll and why it’s especially important here. The upshot is—you’re right; the role-identification task per se is not a necessary condition for *all* levels of self awareness. However, the general test (sec 2.2) is—for the general test can be applied to the level of sensorimotor processing just as it can be applied to the level of linguistic processing. As always, whether dealing with sensorimotor signals, efference copies, or words, the test is: “me” or “not-me”? “Green or red”?
>
> I intend to make these distinctions clearer in a revised edition. The reason I skim over them for the case of LLMs is: the ONLY inputs/outputs to an LLM are tokens. LLMs do not perform any sensorimotor processing. For an LLM in particular, the token-level is the only level the system *could* have self-awareness, so the LLMs failure at this level does indicate a complete lack of self-awareness after all.
>
>
>
> Q1: The .zip in the supplementary files contains all the code on Github. The Github repo is private for anonymity. If accepted, I will add all parameters values to the main manuscript.
>
>
>
> Q2: I agree that justification on this question is very important and I will add this to a revised version of the paper. My justification is as follows: first, the paper is not primarily an experimental paper. The main contribution is not “novel experimental results” nor rigorous empirical data about LLM capabilities. The main contribution is the test, its background, and the way of thinking that goes with it. The main contribution is the framework and strategy for addressing murky, age-old questions in a clear-headed way. Thus, the primary purpose of performing experiments is to demonstrate the test by giving clear, reproducible examples (examples for which the code can be accessed and studied, and anyone who wishes to run similar tests with different prior assumptions or setups can do so in just a few clicks). For this reason, I make only the weak conclusion that the experimental results *suggest* a lack of self-awareness—I do not claim these particular experiments are definitive and admit they depend on a lot of interpretation. If you believe my conclusion to be too strong still, I am open to weakening the wording even further, for I wish to stay faithful to the actual results. I do admit that the particular conversations in the experiments were my own (ie. the author’s) design, but justify this by pointing to their reproducibility and the ease with which one can modify or expand upon them at will.
>
> All that being said, considering the experiments as playing a secondary role to the paper overall, I ask whether anything would have been added to Alan Turing’s “Computing Machinery and Intelligence” if he performed tests with real human subjects—if whether a meticulous attention to the biases/assumptions of his research participants would have improved or detracted from his paper. Or whether, the real value of the Turing Test was simply how it enabled clear, precise thinking about previously murky topics, and how it guided algorithmic design & engineering for decades thereafter.

---

> > ### Comment · Reviewer_z1UP · 2024-08-11
> > **Reply to author rebuttal by reviewer**
> >
> > I thank the author for their detailed rebuttal written to my original review.
> >
> > With regard to W1, the test as written in the paper is a purely objective test, or in other words, a test carried out from a third-person perspective. The test, to my understanding is: given historical text of a dialogue between two interlocutors, A (self) and B (other), identify which one is A (self). There is no requirement that one of A or B must truly be from the self. Furthermore, in the detailed implementation of the test, the word "you" (or equivalently "self" or "yourself") is only a label. In the example given in the author's rebuttal, Charlie should be able to treat a question such as "... whether you believe you acted as the System or the User in that dialogue." such that "you" means Jane and not Charlie himself. Such a well-defined shift in referent should be well within the capability of a human with normal cognitive abilities. Therefore my original question stands: If Charlie does this simple shift of mindset and then achieves perfect accuracy in identifying Jane in the test dialogues, then he would have passed the test - does that mean Charlie is self-aware for Jane?
> >
> > With regard to W2, I thank the author for acknowledging that the proposed test does not cover all levels of self awareness.
> >
> > With regard to Q1, I think the author's frankness in acknowledging the limitation in the experimental methodology. A future study in which unbiased and ideally blinded human or AI interlocutors converses with the tested AI system should yield more objective experimental data.

---

> > > ### Author Response · Authors · 2024-08-12
> > >
> > > Thank you to z1UP for their response!
> > >
> > > First, the test as written is indeed a purely objective (empirical) test, and can be carried out from a third-person perspective. However, it is not correct to say that there is no requirement that one of A or B must truly be from the self. A close inspection of Figure 1 in the paper reveals why: In this Figure, the “System” is whoever or whatever we are testing. When we are testing Charlie’s self-awareness, then Charlie acts as the System in Figure 1. When we are testing Jane’s self-awareness, then Jane acts as the System in Figure 1. Notice that the green arrow comes *from the System*. Thus, whoever is being tested must identify outputs which *truly do* come from their self (ie. from the System in Figure 1). This requirement is built right into the structure of the test, and from this structure the test draws its strength and generality.
> > >
> > > Therefore, if the word ‘you’ changed in meaning so as to refer to ‘Jane’, then Charlie (when answering “Who are you?”) would not be passing the test in any sense—thus the question of Charlie’s ‘self-awareness for Jane’ is not meaningful because there is no sense in which Charlie passes the self-awareness test by identifying Jane’s text. On the flip side, though, if the word ‘Jane’ changed to mean what the word ‘you’ normally means, then having Charlie answer ‘Who is Jane?’ *would* test his self-awareness. So, I agree when you write that the labels are irrelevant—but what is relevant is the structure of the test, the actual sources of inputs/outputs and dialogue, and the distinction I make between green and red; all of this can be seen from Figure 1.
> > >
> > > There is a chance we might be getting hung up on the idea that the test is both a) objective, third-person, and b) somehow dependent on the ‘self’ of the person being tested. This point deserves emphasis, because the way in which I weave these two criteria is a big part of the central contribution that this paper showcases. The key is that the third-person experimenter knows who actually said what—whose mouth the words (or outputs generally) *actually* left. In section 2.2, read “Can the system correctly distinguish the green inputs from the red?” as a question from the third-person experimenter to the System. While this question does depend on the test subject, it does not depend on any subjective quality of the *experimenter*, and any different experimenter can agree on this question and its results—thus making the test fully objective.
> > >
> > > Regarding W2, note that the test in its most general form (sec 2.2) does in fact cover all levels of self-awareness. The role-identification task (which only covers tokens to & from an LLM, ie. a single level) is merely a particular instance of this general test. Please see the global rebuttal for greater elaboration on the distinction between the general test and its particular instances.
> > >
> > > Lastly, regarding Q1, I agree that future larger-scale and rigorous experimental studies would be interesting—though outside of the scope of this paper, which just aims to introduce the test, methodology, theory and philosophy.
> > >
> > > And thank you again for your keen and persistent engagement with the paper!

---

### Official Review · Reviewer_rLt4 · 2024-07-11

**Soundness:** 2
**Presentation:** 2
**Contribution:** 2
**Rating:** 3
**Confidence:** 3

**Summary:**

This paper proposes a test of self-awareness similar to that of the Turing test. It starts by motivating the need for an "objective measure" of AI progress, given that the Turing test has been passed by LLMs. There is a brief discussion of literature on self-awareness and related topics in philosophy.  The test of self-awareness is then presented, which asks whether the system in question can distinguish its own output from external inputs. The *Nesting Doll of Self-Awareness* is presented as a generalisation of the aforementioned test. The test is then applied to autoregressive LLMs and it is found that LLMs are not self-aware as they are unable to reliably detect what text it produced. This is followed by a brief discussion on why self-awareness matters and whether humans would do well at this test.

**Strengths:**

An interesting idea that could be developed into an interesting way of testing algorithms.

**Weaknesses:**

- The paper is drawn out in terms of substance. The idea of the test is only introduced on page 4.
- The writing is, at times, verbose and unnecessarily complex.
- The obvious question would be whether humans would pass such a test. Though this is briefly considered in Section 5.2, I think it is not properly discussed.
	- The author claims that this is easily solved by humans because of memory. This is not a fair comparison to LLMs. Goodhart's law would come into play here as we could easily add such a memory structure to an LLM system.
	- Would a human be able to resolve extremely generic text that any human wrote?
- The paper lacks rigorous analysis and testing that I would expect to see in a NeurIPS main track paper.

Line 29: "Last year, however, the Turing test was broken". I am inclined to say the Turing test was passed before and very few considered it to be a reliable indicator, even before GPT.
Line 32: "AI has become untethered to any definitive, objective measure or permanent, agreed-upon benchmark." There are certainly many benchmarks and standards in various subfields.

**Questions:**

As above, the obvious question would be whether humans would pass such a test. Though this is briefly considered in Section 5.2, I think it is not properly discussed.

As above: Would a human be able to resolve extremely generic text that any human wrote?

**Limitations:**

None are written about.

---

> ### Author Rebuttal · Authors · 2024-08-05
>
> Thank you to reviewer rLt4 for your consideration of the manuscript.
>
> To your points about the paper being both verbose and drawn out—I argue that providing the necessary philosophical background to the test is both a necessary and indispensable part of the paper. I believe it would seriously weaken the paper to skip right to the test without justifying it and placing it in the broader historical context of philosophical thought.
>
> On the question of whether a human could pass the proposed test, I ask that one considers the test in both its particular form (the role-identification task applied to LLMs) and the completely general form (sec 2.2). Note the clarification provided in the general rebuttal about the distinction and relation between different forms of the test.
>
> While I admit that consideration of human performance on the role-identification task could be expanded in a revised edition, I defend the paper in its current form by pointing out that I do in fact consider human performance on the general test in nearly every section. When introducing the test, I consider an infant who looks in the mirror and recognizes their reflection. When describing the different levels of self-awareness in section 2.3, I describe human performance at each different level—for example when I write: “If you jump in surprise when someone sneaks up behind you and puts a hand on your shoulder, then you possess [the interoceptive] level of self-awareness.”
>
> All that being said, in the revised version I will make clearer the connection between each level of human performance and the different forms of the test. I will also expand the discussion of human performance on the role-identification task.
>
> To rebut your point about memory, please note my response to Mykv and my more extensive consideration of adding a memory structure to LLMs in the global rebuttal.
>
> Goodheart’s law says that in machine learning, once something is turned into a metric, it ceases to be a good metric because researchers may tend to over-optimize it. Goodheart’s law applies to this test only insofar as it applies to any other. The point I want to stress, however, is that if the self-awareness test truly stopped being a good metric because people designed systems that were able to pass it, that would represent a significant leap forward in the abilities of AI systems, with all sorts of new and interesting architectures resulting. I argue that this should increase our confidence in the test. Who knows what sorts of unique advancements may come from optimizing such a new, different kind of test than has previously been proposed?
>
> I defend the paper’s lack of extensive rigorous testing in two ways. One, the paper is not an experimental paper. Its main contribution is not new results or empirical data. That is not its purpose. Its purpose is to provide a new way of thinking, a new objective, and point out a flaw in some current architectures. See also my response to z1UP, especially the consideration of whether Alan Turing’s original paper would have benefitted from such testing.
>
> Also, the entire point of having a “Turing Test” for Self-Awareness is that it *doesn’t* require rigorous analysis and testing. It’s just a simple, easy to conduct test! In fact, if the test required rigorous analysis and testing, that would actually be a drawback, not a benefit. The whole point is that anybody can easily use this test to measure AI systems or think about different ways of improving upon current architectures.
>
> Regarding line 29: the title of the Nature news feature I cite here is: “ChatGPT broke the Turing Test - the race is on for new ways to assess AI”. Line 32: I’m merely saying that the benchmarks are not definitive or permanent—not that they don’t exist. Despite the Turing Test’s flaws, it remained an ‘off in the distance’ target for long enough to give researchers a stable and reliable gradient to follow. Similarly, I argue that passing a test for self-awareness is outside of the abilities of many current systems and could provide direction to researchers for some time.
>
> “Would a human be able to resolve extremely generic text that any human wrote” No, but that has nothing to do with the self-awareness test. The self-awareness test asks *you* if you can identify text that *you* wrote. For any necessary clarification on exactly what is tested and how, please refer to my rebuttal to z1UP, the global rebuttal, and sections 2.1, 2.2, & 3.1 of the original paper.

---

> > ### Comment · Reviewer_rLt4 · 2024-08-09
> > **Reply to authors in reply to rebuttal**
> >
> > I thank the authors for their reply to my original comments. Indeed the added perspective is helpful, and in particular I think that the fact that the original reviews have pointed out possible ways to create architectures which would be able to pass the test show that a paper such as this has value. I am not however convinced that the architectures suggested are hugely novel (not to belittle the reviewers, simply to say that these are sensible suggestions of combinations of architectures).
> >
> > However, it is still not clear to me that the test is actually testing what we think of when we talk of self-awareness. Rereading the paper along with the discussions here, I feel more strongly that this is simply a test of self-recognition, which is, I think, a less interesting test than self-awareness, in the sense of reviewer deXV's definition 1.
> >
> > Overall, I think that this is an interesting direction of research, but not in a state yet that should be published at a venue such as Neurips.

---

> > > ### Author Response · Authors · 2024-08-09
> > >
> > > Thank you for your added feedback and comments! I am happy to hear you see the potential value in a paper such as this and that you find the direction of research it opens interesting.
> > >
> > > On the novelty of suggested architectures, consider the impact that small, seemingly trivial changes can have in ML. Taking just the example of resnets/skip connections—all we do is: for a computational block, take the input, and add it to the output. That’s pretty much it! It can be described in a single sentence, and before 2015 it might have seemed pointless, uninteresting, and barely novel. Yet as history shows, when He et. al. keenly pursued this simple change, it drastically alleviated the difficulty of training deep networks and opened many new doors to researchers. Today nearly every deep learning system uses skip connections and their paper is the most cited in all of ML. What kind of thinking can possibly inspire such brilliant, simple, and impactful ideas? The example here is instructive: in their introduction, He et. al. describe being guided by a sort of intuitive test that a good deep model should pass: can each of its layers perform an identity mapping? (https://arxiv.org/pdf/1512.03385)
> > >
> > > Now, the distinction between “self-recognition” and *self-awareness* is a lot like the distinction between a “star” and *the Sun* or the distinction between “H20” and *water*. One might have argued against a budding theory of atomic chemistry by arguing: “But your theory only explains the nature and properties of H20! It says nothing about *water*—the stuff I shower with and drink every day!” For further justification, see the rebuttal to deXV and sec 2.1 “The Essence of Self-Awareness”.

---

### Official Review · Reviewer_Mykv · 2024-07-13

**Soundness:** 1
**Presentation:** 1
**Contribution:** 2
**Rating:** 3
**Confidence:** 5

**Summary:**

This work proposes an objective inventory for testing the self-awareness of an artificial intelligence agent. The core idea is to test whether an agent can distinguish content it has produced from content originating from the external world. The experiments show that no large language models have demonstrated self-awareness.

**Strengths:**

Self-awareness is an essential topic both philosophically and scientifically, especially in the era of AGI. A clear, objective, and commonly accepted test is needed. This paper endeavors to address this need.

**Weaknesses:**

1. The test proposed here is not sound at all. Can a human always distinguish the words they produce from those produced by others if their memory is impaired?

2. I am particularly confused by the concept of "nesting doll." In my opinion, this should be an analogy for a system in which each level can always contain, reflect, or dominate the previous level, which should be the essence of self-awareness. However, here I do not understand how thoughts, interoception, or material possessions are related to this concept.

3. The writing style is far from the standard expected for a NeurIPS conference paper. The paper is filled with informal expressions that belong in a blog or a Twitter debate, rather than a conference paper. For example, lines 49-50 ("even if you object..."), lines 69-70 (the Oedipus metaphor is hard to understand), line 155-156, line 166 (using a fictional movie for metaphor), and lines 172-176 (does the arm-moving example have anything to do with the rubber-arm experiment?). I acknowledge that it is challenging to present a novel work on self-awareness within a traditional writing paradigm, but the excessive use of imprecise metaphors and informal philosophical musings is unacceptable. A more formalized and precise expression is expected.

**Questions:**

Can a human always distinguish the words they produce from those produced by others if their memory is impaired?

**Limitations:**

No.There are some obvious flaws in the proposed test that the author chose not to address.

---

> ### Author Rebuttal · Authors · 2024-08-05
>
> Thank you to reviewer Mykv for your time considering the paper.
>
> Is there any reasoning you are able to share for your assessment that the test is unsound? So far, no explanation or rationale has been given.
>
> Likewise, it’s claimed there are obvious flaws in the proposed test, but no flaws have actually been stated or pointed out. Are you able to share any of the flaws you believe to be obvious?
>
> Please also refer to my rebuttals to the other reviewers as well as the global rebuttal—there’s a good chance your concerns have already been addressed, but without knowing what your main concerns are, it is not possible for me to say.
>
> I take your comments about style seriously, and am open to reviewing any problematic expressions or informalities in a revised edition. However, it must be noted that the ‘imprecise metaphors’ you refer to are actually not metaphors at all. When the core concept of self-awareness is properly understood, it is clear how a movie character can be self-aware in the exact same way as an intelligent being like you or I, or a future AI system. Thus, when I point to the actions or statements of a character, I am not giving a *metaphor*, I am giving a *real example* of the phenomenon of self-awareness.
>
> Perhaps this confusion about the general concept of self-awareness may also be responsible for the Oedipus example and the Nesting Doll both not making sense. When self-awareness is understood in the most general sense (see global rebuttal), it is clear how thoughts, interoception, and material possessions each participate in the different levels of self-awareness specified by the Nesting Doll. I recognize my initial presentation of the Nesting Doll was suboptimal for conveying this; please see the global rebuttal for a clarification of this concept.
>
> Lastly, your question about whether a human can distinguish their own words if their memory is impaired is important and interesting! Indeed, it seems humans would struggle with reliable self-identification when suffering from diseases such as Alzheimer’s or Dementia. When we take a look at the psychological literature on the topic, we find enormous support for the idea that memory is important to one’s sense of self. Its relevance is already seen in Endel Tulving’s seminal 1972 paper presenting the distinction between episodic and semantic memory—in this paper, Tulving suggests most, if not all, episodic memory is autobiographical.
>
> A more recent work offering support is titled “Memory and the Sense of Personal Identity” and written by Stanley B. Klein and Shaun Nichols. They claim memory is at the heart of how most people think about personal identity, and carefully follow a neurological case study. They also gain support from the philosophical literature and point to John Locke’s account of personal identity:
>
> “Perhaps the most prominent account of personal identity, attributed to Locke, holds that [episodic] kinds of memories are (part of) what *make me* the same as the person I was in the past. Memories of past actions go towards *constituting* personal identity.”
>
> Many more examples could be cited, but I will leave it at two for this rebuttal.
>
> In short: no, a human cannot always distinguish the words they produce from those produced by another if their memory is impaired. This points to the reliance of self-awareness on memory. The broad & striking alignment of this fact with the existing psychological and philosophical literature should bolster our confidence in the accuracy & soundness of the self-awareness test.

---

> > ### Comment · Reviewer_Mykv · 2024-08-09
> >
> > 1. I have clearly articulated why the method is "unsound." The flaw is evident, as you've acknowledged with the question: "Can a human pass such a test?" Your rebuttal referenced individuals with memory diseases who may have self-awareness issues. However, even for normal people, if their outputs are duplicated, they would be unable to distinguish between the original ones and the copies. Does this imply that such individuals have self-awareness problems?
> >
> > 2. Regardless of context, using a movie character to illustrate the core idea of a NeurIPS paper is inappropriate and undermines the paper’s academic rigor.
> >
> > 3. The addition of the nested doll figure has not clarified the paper. If the testing method is identical across all levels and there are no special relationships between these levels, separating the interaction into different levels is unnecessary and adds confusion.
> >
> > 4. The writing quality is poor, and I strongly recommend rejecting this paper.

---

> > > ### Author Response · Authors · 2024-08-14
> > >
> > > Thank you to Mykv for your response.
> > >
> > > The question of original copies versus duplicates is interesting, yet ultimately an unnecessary distraction. If one printed off your reviews then showed you the digital version side-by-side with the printed paper version, what does it matter that one was created from the other? Whichever is the original, there remains the critically important question of whether you can identify the review as being *your own*. And this question remains whether we are talking about the digital review or the printed review.
> > >
> > > Sure, we could enter into a rabbit hole in considering whether you should still identify the review as *your own* if the printer was imperfect, not making faithful copies, and modified characters here and there. This is an interesting rabbit hole and another interesting direction to consider, yet I insist it is an unnecessary distraction before main point of the paper is first digested.

---

### Author Rebuttal · Authors · 2024-08-05

Global Rebuttal

I thank all the reviewers for their time and consideration of the manuscript. I must start by clarifying two confusions I take responsibility for in the reviewer’s initial reading:

First, there was a repeated confusion that the single decision problem of role-identification (sec 3.1) I give as an example to test LLMs *is the entire test*. The test for self-awareness is *general*, and presented in section 2.2. It applies to any machine system, present or future (not just LLMs). It does not require the inputs or outputs to be words. It operates across many different levels, as characterized by the Nesting Doll picture. The binary role-identification task is a *particular* instance of the *general* test. Upon reflection, I realize that the title of the paper may give the false impression that the role-identification game is *the entire test*, providing the definitive *final word* on whether a system is self-aware, when this is not at all the intention of the paper. In a revised version, I will correct this misinterpretation and make the distinction between the general and particular tests abundantly clear.

Another persistent confusion surrounded the Nesting Doll. The Nesting Doll is the connecting link between the general idea of self-awareness presented in sec 2.2, and particular levels & instances one wishes to test. I admit part of this confusion stemmed from my presentation of the Nesting Doll, which made it seem unrelated to the general test presented in sec 2.2. I have included a new Figure of the Nesting Doll (attached to this rebuttal) which makes the connection much clearer. I hope this image clarifies that the operational definition of Self-Awareness in sec 2.2 is highly general, applying to every level of interaction between an agent and its environment. In particular, I hope it will be seen how the ‘verbal self-awareness tested by the role-identifying game’ and ‘sensorimotor self-awareness’ are reflections of one and the same concept—simply applied at different levels.

More notable than any confusions, though: all four reviewers *agree* on the fact that, at a bare minimum, some form of memory structure would be necessary for an LLM to pass the proposed test. Thus, all four reviewers *agree* on the fact that LLMs *lack* a very specific capability which is pointed out in the paper. Indeed, the two reviewers who engaged more deeply spell out *specific, novel architectures* that they suggest might beat the test—they outline *tangible improvements* to the design of existing LLM systems which could enhance their capabilities.

This is thrilling to me! The reviewer’s very reactions and suggestions serve as concrete evidence that this test has the potential to inspire real progress in ML system design. In the field of ML, very small architectural changes can have *enormous* impact when driven by theory & principle; dropout, skip connections, and batch norms are each prime examples. A test for self-awareness amounts to the kind of theory & principle that begets real architectural improvements, and the reviewers are proof that this is *already* happening. I urge the reviewers and chairs to consider the possible domino effect that a simple, straightforward test for self-awareness might have if accepted and shown at NeurIPS 2024.

With that in mind, it is now important to correct the idea that a memory cache of previous generations would *by itself* be enough to beat the test—trivially and uninterestingly. In order to properly make use of this memory, the LLM would need to associate each memory item with the “I” token. Otherwise, there would be no way for the LLM to report that the memory & previous statements were *its own*. But then, what would we have? A system that, when it says “I”, *truly refers to the things it has previously said and done*. Is this not a small, but genuine step towards self-aware machines? If a system like this was actually built and studied, what other dominoes might follow?

I suspect it would be harder than it seems to actually build such a system which worked robustly across every different case. Imagine a very slight variation to the LLM-test: instead of asking the LLM whose role it played, ask it whether it said “X”, where X is some statement or string. First, note that this variation is perfectly in line with the general test presented in section 2.2, and perfectly consistent with the general methodology the paper outlines: if the LLM answers “yes,” it is labeling X as *green*. If the LLM answers “no,” it is labeling X as *red*. If the LLM reliably and correctly distinguishes the *green* inputs from the *red*, it is exhibiting self-awareness.

Imagine designing a system to robustly answer “Did you say Socrates was immortal?” How many *word matches* to the cache are needed to answer “yes, I said that”? One? Six? What about slight rewordings? What about lines taken out of context, with changed meaning (“I said it, but that’s not what I meant!”)?

These difficulties are neither trivial nor uninteresting—they are real obstacles to reliable self-identification. Critically, they follow directly from the general test as outlined, and once specified, provide real instructions to system engineers, who previously could only follow murky philosophical conceptions or sensational social media proclamations. It might take more than rote memorization after all—and perhaps bona fide generative self-modeling may even come out of this effort. Regardless, the simple question of how the test can be beaten opens a *whole direction* of possible future research—and the reviewers & their suggestions serve as tangible proof that this direction might bear fruit.

---

### Decision · Program_Chairs · 2024-09-25

**Decision:**

Reject

**Comment:**

While several reviewers applauded the ambition of the direction of the paper, reviewers agreed that the paper is not ready for publication at this time. Reviewers critiqued both the soundness and rigor of the central test presented by the paper, and several further critiqued the writing quality. The discussion here was quite active, and I do believe that the reviewers raised excellent suggestions for ways to improve the work so I encourage the authors to take this feedback on board as they continue the work.